# Fronts and waves of actin polymerization in a bistability-based mechanism of circular dorsal ruffles

Erik Bernitt[1,2,3], Hans-Günther Döbereiner[3], Nir S. Gov[1] & Arik Yochelis[2]

During macropinocytosis, cells remodel their morphologies for the uptake of extracellular matter. This endocytotic mechanism relies on the collapse and closure of precursory structures, which are propagating actin-based, ring-shaped vertical undulations at the dorsal (top) cell membrane, a.k.a. circular dorsal ruffles (CDRs). As such, CDRs are essential to a range of vital and pathogenic processes alike. Here we show, based on both experimental data and theoretical analysis, that CDRs are propagating fronts of actin polymerization in a bistable system. The theory relies on a novel mass-conserving reaction–diffusion model, which associates the expansion and contraction of waves to distinct counter-propagating front solutions. Moreover, the model predicts that under a change in parameters (for example, biochemical conditions) CDRs may be pinned and fluctuate near the cell boundary or exhibit complex spiral wave dynamics due to a wave instability. We observe both phenomena also in our experiments indicating the conditions for which macropinocytosis is suppressed.

[1] Department of Chemical Physics, Weizmann Institute of Science, Herzl St 234, Rehovot 76100, Israel. [2] Department of Solar Energy and Environmental Physics, Swiss Institute for Dryland Environmental and Energy Research, Blaustein Institutes for Desert Research (BIDR), Ben-Gurion University of the Negev, Sede Boqer Campus, 849900 Midreshet Ben-Gurion, Israel. [3] Institut für Biophysik, Universität Bremen, Otto-Hahn-Allee 1, 28359 Bremen, Germany. Correspondence and requests for materials should be addressed to A.Y. (email: yochelis@bgu.ac.il) or to N.S.G. (email: nir.gov@weizmann.ac.il).

Living cells use distinct strategies for the uptake of extracellular matter as the cell membrane is impermeable to most molecules[1]. For the unspecific internalization of solutes one of the respective processes is macropinocytosis in which cells grow actin-based vertical protrusions (ruffles) that collapse or fold back on either themselves or the cell body. The collapse creates enclosed cavities that are then internalized as vesicles (macropinosomes). The extracellular fluid inside of macropinosomes contains, for example, dissolved macromolecules that serve the nutrition of cells[2,3].

Cells maximize the macropinosome volume by organization of ruffles into circular, contracting cup structures[2,4]. During this self-organization process one observes highly dynamic ring-shaped undulations known as 'circular dorsal ruffles' (CDRs)[5–7]. Importantly, although CDRs provide the basis for the formation of macropinocytotic cups, they are not necessarily structures of high vertical extension themselves (compare, for example, the scanning electron micrographs of CDRs in ref. 5, where they appear as shallow undulations, with ref. 4, in which they rather form circular arrays of extended, lamellar protrusions). Thus, the frequently used 'ruffle' terminology, which has historical reasons[8], is somewhat inappropriate in this context. CDRs often initiate in a localized spot on the membrane from which they expand as a ring, which then reverses and contracts centripetally. The cup structure that forms during the contraction finally collapses, which leads to the formation and internalization of one or more macropinosomes[9,10]. Apart from nutrient supply, CDR-mediated macropinocytosis serves to internalize activated receptors that bind, for example, growth factors, and therefore plays a central role in the down-regulation of signalling events[11]. Thus a lack of CDRs presumably facilitates the uncontrolled growth of tumour cells[3,7,11]. On the other hand, CDR-mediated macropinocytosis has been identified as an important mechanism of nutrient uptake, especially in tumour cells[12] and, further, CDRs have been suggested to support cancer cell migration by softening of the cytoskeleton through disruption of stress fibers[7]. Moreover, pathogens, such as salmonella, hepatitis viruses and HIV, are known to hijack CDRs as a 'gate opening' to enter the host cell[3,13,14]. Due to their clear significance to these pathological and physiological processes alike, CDRs have gained a lot of attention from the biological community, leading to a comprehensive proteomic characterization[5–7]. Yet, to date it is unclear how proteins in CDRs self-organize to form these dynamic ring-shaped structures[6,7,15].

It is known nevertheless, that CDRs constitute a type of actin wave, that is, a propagating wavefront of actin polymerization at the dorsal cell side[9,16,17]. Actin waves have been observed at the ventral cell side or the cell periphery before[18–24] and have been studied via reaction–diffusion models in the context of excitable[17,25,26], wave unstable[27] and bistable[28,29] dynamics, as well as in terms of actin-membrane shape feedback[16,30,31]. However, the phenomenology that incorporates the initial expansion of a circular wavefront from a localized initiation spot, eventual contraction, and ultimate collapse of which the

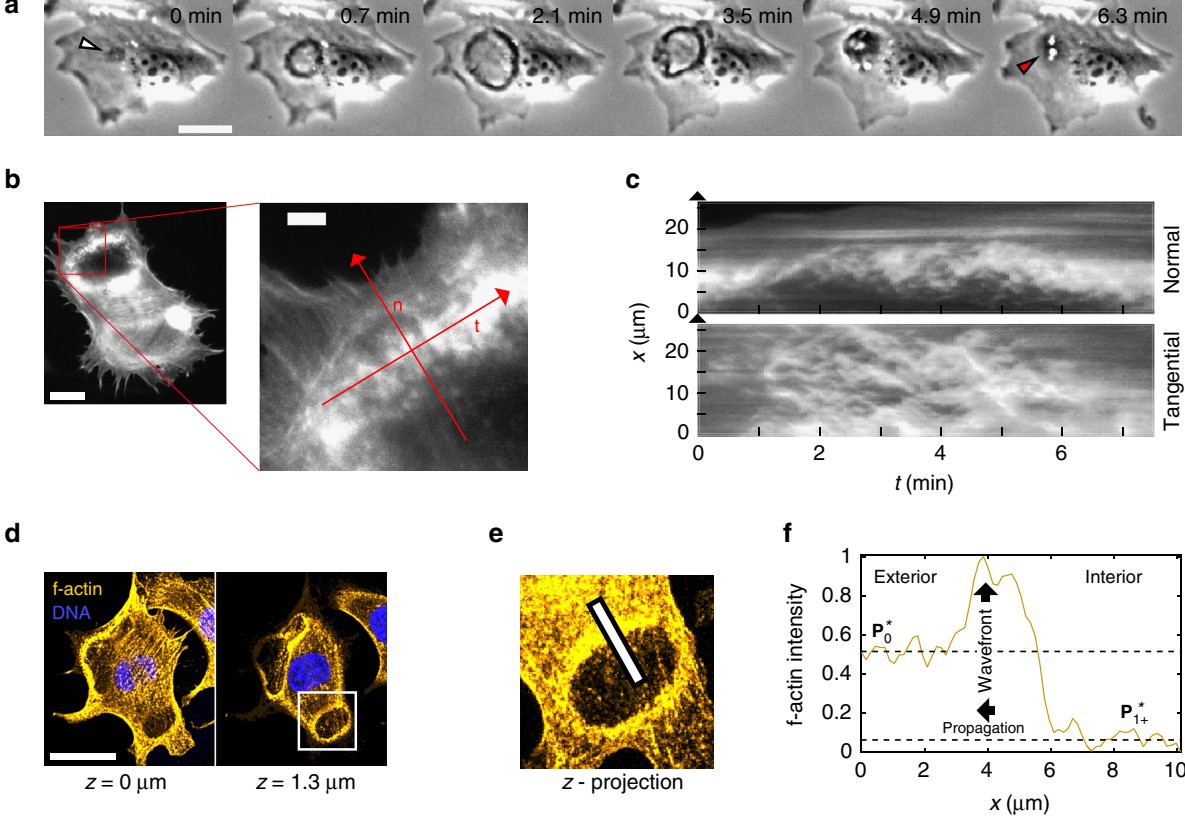

**Figure 1 | Characteristics of CDRs.** (**a**) Time-lapse sequence showing the typical course of a spontaneously formed CDR (scale bar: 25 μm). The white arrow indicates the initiation of the CDR and the red arrow the macropinosomes (appearing as white spots) formed on CDR collapse. (**b**) Living cell stained for f-actin with a close-up view on a CDR wavefront showing its sub-structure of dynamic actin clusters (scale bars: full image 25 μm, close-up 5 μm). (**c**) Kymographs along red lines in normal (n) and tangential (t) direction to the CDR in **b**, highlighting the rapid actin turnover within CDR wavefronts. (**d**) Actin organization of a cell exhibiting two CDRs imaged with confocal fluorescence microscopy in two different z-positions. (scale bar: 25 μm). (**e**) Close-up view of the vertically integrated intensity of the region of interest highlighted with a white rectangle in **d**. (**f**) Profile of fluorescence intensity sampled along a cut through the wavefront (white line in (**e**), length: 10 μm) showing the state of wavefront exterior ($\mathbf{P}_0^*$) and wavefront interior ($\mathbf{P}_{1+}^*$).

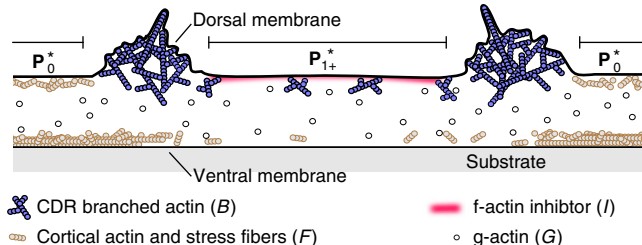

**Figure 2 | Schematic sketch of the distribution of actin and the inhibitor of actin polymerization in a cross section through a CDR.** The regions corresponding to the fixed points of the model equations $P_0^*$ and $P_{1+}^*$ are marked around the CDR location.

latter process underlies the endocytotic function of CDRs, is unique and currently neither understood nor captured (as a whole) by any existing modelling attempt.

Here we utilize confocal fluorescence microscopy to show that CDRs seem to obey bistable kinetics, which manifests in distinct actin densities between the interior and exterior. On this basis, we propose a novel mass conservative reaction–diffusion model that incorporates three actin states. Through analysis of the model equations, we not only reproduce and relate the CDR dynamics to counter-propagating fronts, but show that under certain biochemical conditions CDRs can become pinned and even show complex dynamics that leads to structural loss of the coherent circular wavefront. Both of these scenarios are then also identified in our experiments and found to suppress macropinocytosis. The spatiotemporal dynamics is robustly determined by bistability and by wave instability of one of the two states. Consequently, our rather simple model captures and generalizes many aspects of actin waves that have been modelled separately[20,22,24,26,27,29] ranging from single waves to spatiotemporal chaos-type dynamics. Yet, the model does not attempt to realistically address the final stage of macropinosome formation and engulfing, which requires the description of large scale membrane deformations, which is beyond the scope of this work.

## Results

**Bistable actin self-organization within CDRs.** CDRs typically initiate from localized points on the dorsal membrane and form expanding, ring-shaped structures of vertical undulations that have a clear signature in phase contrast microscopy (first three frames in Fig. 1a). On approach of the wavefront towards the cell nucleus or the cell edge, CDRs usually reverse their propagation direction, contract, and collapse, that is, the ring shrinks and disappears (last three frames in Fig. 1a); at the final collapse stage one or more macropinosomes form (red arrow in last frame in Fig. 1a). Notably, CDRs do not actually reach the cell edge, but rather reverse at a distance comparable to the width of the wavefront (about $5\,\mu m$ to $10\,\mu m$). Additionally, CDRs are not observed to form wave trains[9], in contrast to, for example, actin waves in neutrophils[20].

In the following we analyse the distribution and dynamics of the different self-organized forms of actin via fluorescence microscopy. We differentiate between branched filamentous actin (associated with protrusions such as CDRs), filamentous actin of static cytoskeletal structures (such as stress fibers and the cell cortex) and free monomeric actin[32]. Live cell imaging reveals that CDR wavefronts are composed of micron-sized motile actin clusters with velocities of typically $(0.18 \pm 0.01)\,\mu m\,s^{-1}$ (Fig. 1b), and relatively quick turnover (Fig. 1c and Supplementary Movie

1). Thus, within CDR wavefronts the rates of both actin polymerization and depolymerization are high, leading to rapid exchange between the pool of actin monomers and filaments mediated by the simultaneous activity of actin branching/nucleating proteins and actin depolymerizers/severing proteins.

Three-dimensional imaging of the distribution of f-actin in fixed cells demonstrates that the interior of CDRs is depleted of both actin stress fibers and cortical actin throughout the whole vertical extension of the cell body (Fig. 1d). The actin of the CDR ring that was identified in phase contrast microscopy (Fig. 1a), has only a faint signature at the level of the substrate in confocal imaging. This is best visible close to the cell nucleus, where the cell is relatively thick and thus allows to clearly optically separate it from the ventral cell parts. Hence, in CDRs f-actin is exclusively polymerized at the dorsal cell membrane. This finding is in accordance with previous work showing that N-WASP is the central actin polymerization agent in CDRs, a protein that is active only when bound to membranes[33].

By constructing profiles of fluorescence intensity along the normal direction to the CDR wavefronts (Fig. 1e,f), we find two distinct concentration levels of polymerized actin. In what follows, we refer to these as $P_0^*$ (exterior) and $P_{1+}^*$ (interior), respectively, as shown in Fig. 1f. In between the exterior and the interior we find the CDR wavefront, which is marked as a peak in the concentration of f-actin and corresponds to the dark ring in the phase contrast micrographs in Fig. 1a.

The bistable nature suggests feedback loops between different organizational states of actin. In Fig. 2, we summarize schematically the prominent key features of CDRs that we include in our model. Specifically, we distinguish between three states of actin: (1) branched actin at the dorsal cell membrane, $B$; (2) immobile actin organized into the cell cortex and stress fibers, $F$; (3) globular actin monomers, $G$. In experiments, we have not explicitly stained for the latter but estimate its distribution based on conservation of the total actin. Recent experimental results suggest that the self-organization of actin in CDRs is controlled by inhibition of actin polymerization via pathways incorporating the phospholipid PtdIns(3,4,5)P$_3$ (PIP3) and the protein ARAP1 (see refs 15,34). Both of these components have been shown to localize in CDR interiors.

In the case of ARAP1, the inhibition is in the form of a secondary wavefront that follows the primary wavefront of actin in expanding CDRs, implying a positive feedback from branched actin $B$ on PIP3/ARAP1 (Supplementary Note 1, Supplementary Fig. 1)[15,34]. ARAP1 effectively downregulates the proteins Rac and Cdc42, which are both associated with a branched organization of f-actin[34,35]. Thus, ARAP1 can be considered an inhibitor of actin polymerization. PIP3 is localized in patches within CDR interiors[15], in contrast to its partner PIP2 (PtdIns(4,5)P$_2$), which is uniformly distributed in the entire cell membrane[36]. Both, PIP2 and PIP3 play central roles in actin regulation. PIP2 activates the major promoter of actin polymerization in CDRs, that is, N-WASP, which is not the case for PIP3. Moreover, PIP2 is associated with the dissociation of the capping protein capZ and of the capping/severing protein gelsolin[36,37]. In summary, this implies that in the PIP3-dominated CDR interiors the incorporation of actin into CDRs is disfavoured and is affected by capping/severing activity of gelsolin. The latter also affects stress fibers and the cell cortex, and has been discussed as a major agent of actin remodelling in CDRs[38,39]. Yet, the mechanisms that confine high concentrations of PIP3 and ARAP1 to CDR interiors are not clear presently. A potential role could be played by the vertical membrane undulations of CDRs, as these have been shown experimentally to function as diffusion barriers[40].

We lump together the activity of ARAP1 and PIP3 into a single control complex $I$ that inhibits the polymerization of actin,

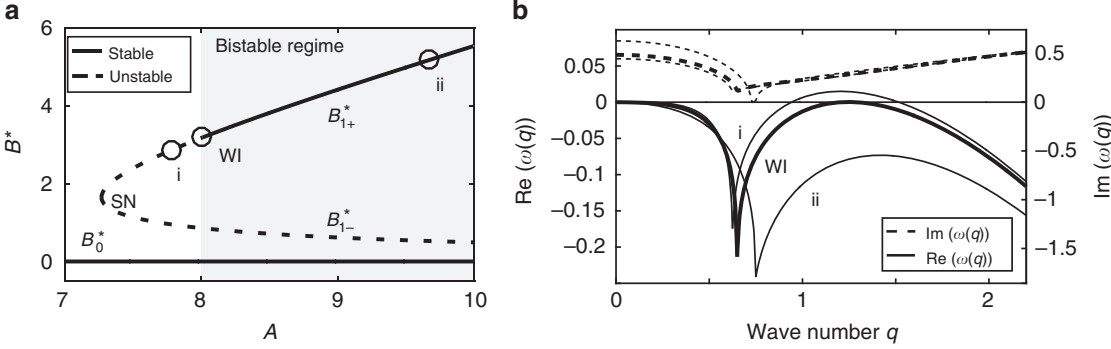

**Figure 3 | Properties of the uniform states (fixed points). (a)** Bifurcation diagram showing the $B$ components of the fixed points $\mathbf{P}_0^*$, $\mathbf{P}_{1+}^*$, and $\mathbf{P}_{1-}^*$ as a function of the total actin concentration $A$. SN and WI mark the locations of the saddle node and Hopf bifurcations. **(b)** Dispersion relation of the linearized system around $\mathbf{P}_{1+}^*$, respectively representing the stability properties at three values of $A$: i (unstable to waves), WI (onset of wave instability), and ii (stable). The wave instability is identified by a positive real part and non-zero imaginary part of $\omega$, whereas negative real parts of $\omega$ correspond to linearly stable uniform states.

and is recruited to the membrane following the CDR wavefront (Fig. 2).

**Mass conserving reaction–diffusion model equations.** The model is two-dimensional, assuming planar and thin cells, as we do not aim to describe features such as the membrane deformations associated with CDRs nor the protein distributions in the vertical direction within the cytosol[29]. Propagation of the CDR, during the expansion and collapse phases, involves mostly minor membrane undulations[5]. Consequently, exclusion of membrane deformations is plausible during these stages. In contrast, at the last stage of vesicle formation, the membrane deformation plays a fundamental role however, it is beyond the scope of this work.

The basic characteristics of the model are (i) positive feedback of branched actin (reaction) and (ii) distinct mobilities (diffusion) of all actin components. Besides these, the remaining terms express standard forms of generation and degradation processes[41]. The model in its dimensionless form reads:

*CDR − incorporated actin*

$$\frac{\partial B}{\partial t} = \overbrace{\frac{B^2 G}{1+I}}^{\text{auto cat.recruitment and polym.}} - \overbrace{B}^{\text{degradation}} + \overbrace{D_b \nabla^2 B}^{\text{diffusion}} . \tag{1}$$

*Stress fibres and cell cortex*

$$\frac{\partial F}{\partial t} = \overbrace{k_{f1} \frac{G}{1+I}}^{\text{polymerization}} - \overbrace{k_{f2} F}^{\text{degradation}} \tag{2}$$

*Actin monomers*

$$\frac{\partial G}{\partial t} = \overbrace{-\frac{B^2 G}{1+I} + B - k_{f1} \frac{G}{1+I} + k_{f2} F}^{\text{conservation}} + \overbrace{\nabla^2 G}^{\text{diffusion}} \tag{3}$$

*Actin inhibitor*

$$\frac{\partial I}{\partial t} = \overbrace{k_{i1} B}^{\text{activation}} - \overbrace{k_{i2} I}^{\text{degradation}} + \overbrace{D_i \nabla^2 I}^{\text{diffusion}} \tag{4}$$

In (1), polymerization of branched meshworks is attributed to a positive feedback, as it is an autocatalytic process due to the increasing availability of filament ends caused by f-actin branching[42]. Moreover, the proteins that cause *de novo* formation of actin filaments, namely the Hem groups of

members of the WASP/Scar family (such as N-WASP, the major nucleation factor in CDRs), exhibit positive feedback loops leading to their own activation via cooperative binding to the cell membrane[20]. The positive feedback in the first term of (1) can also arise due to curved nucleators of actin polymerization that get concentrated at the undulated membrane of the CDR[16]. Since $I$ is an inhibitor of actin polymerization rather than a degrading complex, the most natural choice to describe its effect on the polymerization reaction is via a simple rational function, akin to a Michaelis–Menten type term, that accounts for convergence to zero polymerization at high inhibitor concentration without allowing for negative polymerization rates[41]. Stress fibers in proximity to the dorsal membrane are similarly affected, and therefore also the polymerization reaction of $F$ is inhibited by $I$. In addition, $k_{f1}$ describes the rate of actin polymerizing into stress fibers and the cortical network, $k_{i1}$ the activation rate of the inhibitor $I$ by $B$, whereas $k_{f2}$ and $k_{i2}$ are the respective degradation rates. In the context of mobility, the cell cortex and stress fibers $F$ are immobile, whereas actin monomers $G$, branched actin filaments $B$, and the inhibitor $I$ are diffusive. We non-dimensionalized our model such that the diffusion of $G$ is scaled to unity. Compared to $G$, which is free to diffuse in three dimensions, $B$ and $I$ diffuse only at or in the two-dimensional dorsal cell membrane and thus have correspondingly small diffusion constants of $D_b$, $D_i < 1$. Compared to a system with only one slowly diffusing species, the main consequence of incorporation of a second slowly diffusing species will be a mere smearing of the profiles of front solutions, but no qualitative changes of the dynamics. To keep the number of free parameters small, we thus neglect $D_i$ in the following. The parameter values are given in the Materials and Methods section. The reaction kinetics of the model conserves the total mass of actin. The model version with physical units is given in Supplementary Note 2.

**Bistability.** Neglecting the temporal and spatial derivatives, we find that (1)–(4) admit three uniform solutions (fixed points, denoted by asterisks) of which two are stable ($\mathbf{P}_0^*$ and $\mathbf{P}_{1+}^*$) and one is unstable ($\mathbf{P}_{1-}^*$). The solutions $\mathbf{P}_{1\pm}^*$ appear in a saddle node (SN) bifurcation under variation of the total actin concentration $A = B_0 + G_0 + F_0 = B_{1\pm} + G_{1\pm} + F_{1\pm}$ (Fig. 3a), the latter being a useful control parameter also from a biological perspective. The explicit forms of the fixed points are given in Supplementary Note 3.

In the CDR-free basal state, all available actin is either incorporated into stress fibers and the cell cortex or exists in free

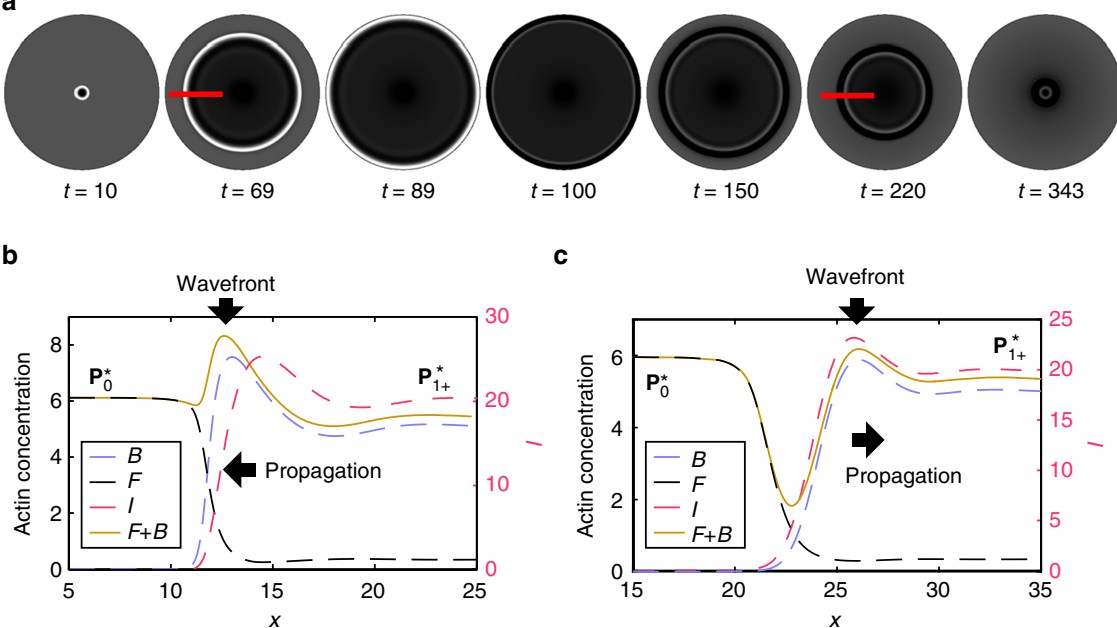

**Figure 4 | Reproduction of physiological-like CDR dynamics.** (**a**) Numerical solution of the model showing an expanding wavefront that reverses propagation direction at the boundary and closes back to a point. The results are plotted in terms of the experimental observable $F + B$. (**b** and **c**) Cross sections in normal direction to the expanding/contracting wavefronts at $t = 69$ and $t = 220$, respectively. The peak in the $B$ field corresponds to the wavefront of CDRs. Note the pronounced depletion of $F$ in the area surrounded by the wavefront.

monomeric form. Thus, the natural fixed point of the system is $\mathbf{P}_0^* := (B_0^* = 0, F_0^*, G_0^*, I_0^* = 0)$. We will show shortly that the phenomenon of CDRs corresponds to the transition of the system from state $\mathbf{P}_0^*$ to $\mathbf{P}_{1+}^*$ where the latter is the state of CDR interiors.

**Linear analysis and wave instability onset.** Next we assume an infinite one-dimensional domain and perform a linear stability analysis to infinitesimal perturbations about $\mathbf{P}_0^*$ and $\mathbf{P}_{1+}^*$:

$$\mathbf{P} - \mathbf{P}_{0,1+}^* \propto e^{\omega t + iqx} + \text{complex conjugate}, \quad (5)$$

where $\mathbf{P} := (B, F, G, I)$ and $\omega$ is the growth rate of perturbations characterized by wavenumbers $q$[43]. Inserting (5) into (1)–(4) and collecting terms up to the linear order, we find via the dispersion relation $\omega(q)$ that $\mathbf{P}_{1+}^*$ can loose stability to waves (a.k.a. finite wavenumber Hopf bifurcation). The instability requires three fields of which at least two need to exhibit diffusion[44–46]. We believe that among the three diffusing quantities, diffusion of $I$ can be neglected as a first approximation. Nevertheless, we have verified that the results and conclusions indeed persist in presence of diffusion in $I$. In fact the addition of diffusion in $I$ results in very minor quantitative modifications. Figure 3b shows the dispersion relation about the bifurcation onset $A = A_{WI}$. A positive real part of $\omega(q)$ is identified with an instability of the respective $q$ while the imaginary part indicates temporal oscillations that eventually result in traveling waves. Thus, for $A > A_{WI}$ the system is bistable. We note that a similar instability has been indeed conjectured as a plausible mechanism for actin waves although in the absence of bistability[27].

**Counter propagating fronts.** Having obtained the bistable regime, we first reproduce the phenomenon of front reversal (Fig. 1a). We use a spatially localized axisymmetric perturbation embedded in $\mathbf{P}_0^*$. From a biological perspective such a perturbation can either be the result of a stochastic event in one of the regulatory pathways, or be caused by the specific binding

of extracellular signalling molecules, such as growth factors, to transmembrane receptors[6,7]. For consistency with the experimental situation, we use Dirichlet boundary conditions by keeping the values of all fields in the state $\mathbf{P}_0^*$ at the domain edges. The localized initial perturbation is in the $B$ field and leads to formation of a propagating front for which the $\mathbf{P}_{1+}^*$ state invades the $\mathbf{P}_0^*$ state, as shown in Fig. 4a. Notably, the $B$ field exhibits a pronounced peak at the front region, which is attributed to the typical ring-shaped zone of high f-actin concentrations in CDRs (compare Fig. 4b and Fig. 1f). The expanding wavefront grows at the expense of the $F$ field, leaving behind a depleted region, which in turn corresponds to the observed loss of stress fibers and cortical actin in CDR interiors (Fig. 1d). The inhibitory field $I$ forms a secondary wavefront behind the peak in $B$, which is in accord with the experimental finding for the field of ARAP1 (Supplementary Note 1, Supplementary Fig. 1)[34], the latter is in fact our main biological inspiration for $I$ (see Supplementary Movie 2 for animated versions of the profiles).

Once the front hits the boundary, it appears to reverse the propagation direction (moving now towards the center), having $\mathbf{P}_0^*$ invading $\mathbf{P}_{1+}^*$ and thus recovering the basal state $\mathbf{P}_0^*$, as shown in Fig. 4a. However, a close examination shows that the two fronts are in fact different, as depicted by their distinct profiles in Fig. 4b,c (Supplementary Note 4, Supplementary Fig. 2). Thus, the bistable nature of our model equations allows also the emergence of wavefront bistability in the form of counter propagating distinct fronts, which is known to occur in non-equilibrium systems exhibiting, for example, the non-equilibrium Ising–Bloch front bifurcation[47–49]. Another intriguing outcome of our numerical results is the co-localization of the peaks of the $B$ and the $I$ field of the back-propagating front (Fig. 4c), which appears to be in agreement with experimental observations (Supplementary Note 1, Supplementary Fig. 1). Eventually the contracting front annihilates, exactly as CDRs do in experiments (compare Figs 1a and 4a). Front collapse after the reversal appears to be insensitive to the domain geometry (Supplementary Note 4, Supplementary Fig. 3). In the

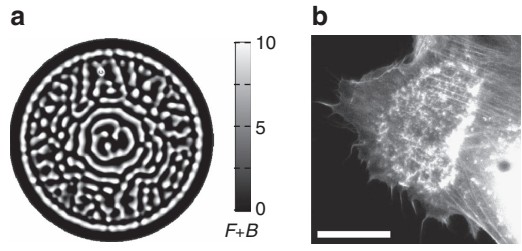

**Figure 5 | Wave instability in CDRs.** (**a**) A typical snapshot of a spatiotemporal numerical solution of the model equations in a regime unstable to waves, showing chaotic dynamics of small waves, surrounded by a coherent wavefront. (**b**) The corresponding situation in a CDR on a real cell (scale bar: 25 μm).

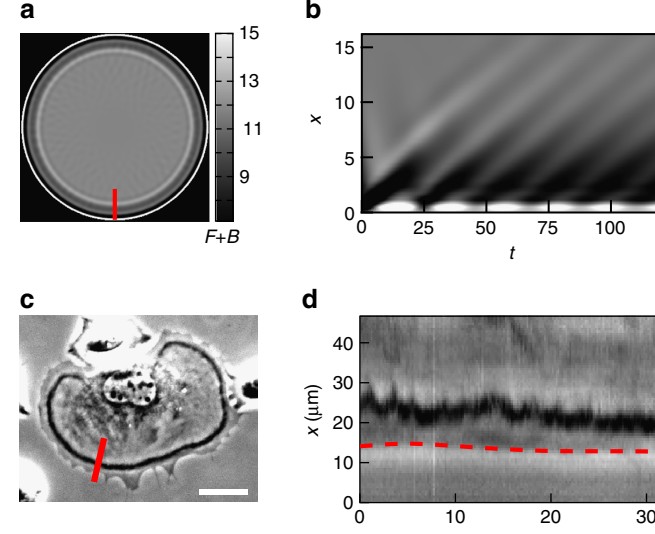

**Figure 6 | Pinned wavefronts in simulations and experiments.** (**a**) Pinned wavefront at the edge of the domain as obtained from direct numerical integration of model equations. (**b**) Space-time plot along red line in **a** showing fluctuations of the wavefront. Note that the colour map was inverted with respect to previous plots for a better visual comparison to experimental data. (**c**) Phase contrast image and the respective (**d**) kymograph showing a typical behaviour of a CDR pinned close to the cell edge (red dashed line). Scale bar: 25 μm.

situation of two colliding circular wavefronts they will fuse and yield one closed front (Supplementary Note 4, Supplementary Fig. 4), which is also in accordance with experimental findings[9].

A detailed comparison between the experimental and theoretical profiles is not given. The reason is that fluorescence imaging of CDRs gives the two-dimensional projection of three-dimensional structures, which can lead to the overestimation of the local actin density in case of significant vertical extension of CDRs. On this basis we currently cannot say with confidence if certain features, such as the pronounced dip in the observable actin in Fig. 4c, are also found in experiments. There are further aspects of CDR dynamics that support our approach. First, the forth and back propagating fronts are distinguished by their speeds: the expanding front is faster than the contracting front (Supplementary Note 4, Supplementary Fig. 2), a feature that agrees with experimental findings[9]. Second, the model also captures the dynamics of artificially induced CDRs. In corresponding experiments, CDRs are reliably invoked by growth factors[4,33]. However, this strategy makes it difficult to control the perturbation shape and to induce a spatially localized stimulation. Indeed, induced CDRs typically lack a pronounced phase of expansion and as such, do not originate from localized points on the membrane[33]. Our model can also reproduce this type of dynamics when the initial conditions are adapted to the situation of growth factor stimulation, that is, the simultaneous perturbation of the entire domain (Supplementary Note 5, Supplementary Fig. 5). Importantly, the robust closure of the actin front in our model does not depend on the initial conditions, in accordance with experiments (compare, for example, Fig. 1a and Supplementary Fig. 5). Finally, both the experimental and numerical results suggest that the expansion and the contraction of the front (the CDR) are based on two regimes of the same nonlinear kinetics and thus do not require any mechanical contractility, contrary to established belief[13] and modelling strategy[29]. The protein typically associated with contractitlity, that is, myosin IIb, indeed was found to be unnecessary for CDR propagation and closure in experiments[16].

**Spiral waves and pinned fronts.** First, we consider a parameter regime $A_{SN} < A < A_{WI}$, for which the uniform state $\mathbf{P}^*_{1+}$ is linearly unstable. Here although the initial perturbation expands at first as a front, the CDR interior in state $\mathbf{P}^*_{1+}$ loses stability to waves that perform chaotic spatiotemporal behaviour including spiraling motion and break-up, as shown in Fig. 5a (Supplementary Movie 3). Such chaotic patterns typically fail to shrink in a ring shape and do not collapse back to a point. However, close to the instability onset the front might still reverse at the boundary and contract. The distance from the instability onset is important here due to the relative time scales of the front

velocity versus the instability growth rate. We indeed find CDRs with similar dynamics also in experiments (Fig. 5b and Supplementary Movie 4). These CDRs fail to form a pronounced ring of polymerized actin, which lowers their ability to undergo efficient macropinocytosis. We note that the proximity to a wave-instability regime may also explain the small motile actin clusters that we identified within CDR wavefronts (Fig. 1b). For $A < A_{SN}$, that is, in the regime where the model has only the single fixed point $\mathbf{P}^*_0$ (Fig. 3a), the system may support excitable pulses. These pulses, which are different from CDRs as they bi-asymptote in space to the same fixed point, may be relevant in the scope of ventral actin waves[20,42]. Currently, this case is not within the scope of our interest and will be discussed elsewhere.

For $A \gg A_{WI}$, we find that wavefronts can get pinned to the boundary where they perform back and forth oscillating motion around a fixed position, as shown in Fig. 6a,b (see also Supplementary Movie 5). The behaviour is related to the absence of a counter propagating front (in this parameter range). A similar regime is also found in experiments. Figure 6c,d, shows a CDR that is pinned at the cell boundary where it initially performs an oscillatory back and forth motion, eventually becoming a stalled wavefront. The overall lifetime of pinned CDRs typically exceeds the normal lifetimes of CDRs considerably (Supplementary Movie 6). Close inspection reveals that the oscillatory motion of the wavefronts at the boundary in Fig. 6b results from periodic emission of wave trains from the boundary, which continue to propagate with weak amplitude towards the domain center. Although stalled CDRs do in fact perform oscillatory motion (Fig. 6d), the experimental data show no evidence of actual wave trains. This might be due to their low amplitude that is possibly beyond experimental detectability. At even higher values of $A$ no oscillations of stalled wavefronts are observed. Naturally, as for CDRs with chaotic wave dynamics, also pinned CDRs fail to undergo efficient macropinocytosis.

For robust closure of CDRs, it is required that cells operate in the bistable regime, but not too far from the instability transition

point $A_{WI}$. In contrast, in the regimes introduced above, cells will not be able to undergo efficient macropinocytosis, and this failure has been associated with cancerous phenotypes[6,7]. Besides the amount of available actin $A$, which was used as a control parameter in Fig. 3a, also the ratio between production and decay of the inhibitor $I$ and the respective ratio for cytoskeletal structures $F$ can serve as control parameters that determine the stability of the system (Supplementary note 3). In biological contexts CDRs are expected to become dysfunctional due to wave instability if cells either exhibit high affinity of actin to form cytoskeletal structures and low rates of their decay or in the corresponding situation for the inhibitory complex. Indeed CDRs that are unstable to waves usually exhibit only weak depletion of cytoskeletal structures (Fig. 5b). Far in the reverse regime, CDRs can become dysfunctional due to wave pinning. Again this is in accord with the experimental observations of vast stress fibre depletion in cells with pinned CDRs[9]. Thus, pinned CDRs might primarily aid cell migration by cell softening, which has implications for cancer cell motility[7].

## Discussion

Using fluorescence imaging data we have shown that cells can locally switch their actin organization between two states and that the wavefronts of CDRs correspond to the transition from one state to the other. From a dynamic perspective, the phenomenology of CDRs can primarily be attributed to distinct counter propagating fronts in a bistable system exhibiting a wavefront bistability, i.e., a behaviour that is typical for the non-equilibrium Bloch fronts[49]. Through formulation of a novel (and rather minimal) reaction–diffusion model, we were able to capture the unique features of CDR dynamics of expansion, contraction and finally a collapse to a single focal point, without any secondary generation of waves. This behaviour is in good agreement with experimental observations. Bistability is a generic property that favours a single, closed wavefront, which reliably closes back to a focal point even on domains of irregular shapes such as living cells. This provides cells with an inherently robust machinery for efficient macropinocytosis. In contrast, cell-substrate actin waves, such as those in neutrophils[20] and *Dictyostelium discoideum*[23,27,28,50], which are involved in cell motility, adhesion and phagocytosis, do not have this requirement[3]. Consequently, those waves do not close back to points[23]. Thus, the insights provided here are important steps towards understanding the differences between actin wave dynamics at the dorsal and ventral cell surfaces. We note that the bistability in our system stems from reaction schemes of proteins that are in or close to the level of direct interaction with actin, which is in contrast to previous works attributing bistability in the actin system to the states of upstream molecular switches, such as Rho-GTPases[25,51], or feedbacks in the actomyosin system[52]. While the model focuses on the biochemical components, it currently ignores the actual growth of the vertical, cup-shaped protrusions[4,13,29,40,53] that may introduce additional details, such as feedback between membrane shape and actin activity[16,29,30]. Also, we leave out of the scope the rarely arising single spiral formations and the periodic spatiotemporal wave patterns that appear only under spatial confinement[9].

Our reaction–diffusion model constitutes a minimal set of equations, in which the wave instability arises on top of the bistability between uniform solutions. Incorporation of wave generation to coexisting counter propagating fronts results in a novel and rich framework of actin dynamics that captures several qualitatively distinct spatiotemporal behaviours: single wavefronts, disordered waves, and pinned states. Thus, besides giving a robust description of the unique features of regular CDR dynamics, we

have also introduced two qualitatively distinct regimes, exhibiting features such as spiral waves and front pinning at boundaries. These two phenomena have been identified experimentally with failure of macropinocytosis and are suggested to play a role in promotion of cancerous phenotypes[6,7].

## Methods

**Cell culture and imaging.** NIH 3T3 (ATCC CRL1658) and NIH 3T3 X2 (ref. 54) fibroblasts were grown under standard conditions in Dulbecco's MEM containing 3.7 NaHCO₃, 4.5 D-Glucose (Biochrom), 100 Penicillin/Streptomycin (PAA), and 10% fetal bovine serum (Biochrom). Cells were mycoplasm free.

Live cell imaging was carried out on a Zeiss Axio Oberver.Z1 equipped with an incubation system at 37 °C and 5% CO₂. A Zeiss Achro Plan $10 \times$ (NA 0.25) and Zeiss Plan Apochromat $40 \times$ (NA 0.95) were used for phase contrast and fluorescence imaging respectively in conjunction with a Zeiss AxioCam MRm camera. No PDGF was added to the cell medium in experiments with live cell imaging (except for Supplementary Fig. 5). Actin was visualized using pLifeAct-TagGFP2 (Ibidi) and Lipofectamin2000 (Invitrogen) transfection.

Confocal imaging of fixed cells was done using a Zeiss LSM 780 equipped with a Plan-Apochromat $63 \times$ (NA 1.4) objective. Actin was stained via Rhodamin/Phalloidin (Biotium) and the nucleus with DAPI (Roche). CDRs initiated via 30 hPDGF-BB (Cell Signaling Technology) in serum-free DMEM. Cells were then fixed 20 min after stimulation using methanol/acetate (1:1).

**Numerical solutions.** Simulations were carried out as a general form problem of partial differential equations in Comsol Multiphysics 5.2. Circular domains of a radius of 50 and a maximal finite element size of 0.1 were chosen. After neglecting diffusion of $I$, the model in dimensionless form has six parameters, of which effectively four govern the dynamic regimes of the system (Supplementary Note 3). Two-dimensional diffusion of $B$ and relatively large particle size relative to the size of g-actin imply a small diffusion constant compared to the diffusion of unity for $G$. For the case of the complex $I$ we expect small diffusion due to relatively large particle size and the coexistence of bound and unbound states to, for example, $F$, that we do not explicitly model in our equations. We choose all kinetic constant to be of similar magnitude relative to each other and use the total actin concentration as a control parameter for the demonstration of the different regimes of the model. The parameters for the simulations were: Fig. 4a: $D_b = 0.12$, $k_{i1} = 2.09$, $k_{i2} = 0.53$, $k_{f1} = 2.05$, $k_{f2} = 1.19$, $A = 9.67$. Figure 5a: $D_b = 0.12$, $k_{i1} = 1.64$, $k_{i2} = 0.30$, $k_{f1} = 2.05$, $k_{f2} = 2.05$, $A = 11.5$. Figure 6a: $D_b = 0.12$, $k_{i1} = 1.64$, $k_{i2} = 0.32$, $k_{f1} = 2.05$, $k_{f2} = 1.71$, $A = 15.5$.

**Data availability.** All data generated or analysed during this study are included in this published article (and its Supplementary Information Files).

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

## Acknowledgements

This work was supported by the Adelis Foundation and the Ministry of National Infrastructures, Energy and Water Resources of Israel, grant nuymber 3-11430 (A.Y.), the German Academic Exchange Service (E.B.) and the Deutsche Forschungsgemeinschaft grant DO 699/4-1 (H.-G.D., E.B.). Further, we thank the Max Planck Institute for Marine Microbiology in Bremen for access to their confocal microscope. N.S.G. is the incumbent of the Lee and William Abramowitz Professorial Chair of Biophysics and this research was made possible in part by the generosity of the Harold Perlman family.

## Author contributions

E.B., N.S.G. and A.Y. designed the research and devised the mathematical model; E.B. and A.Y. conducted the analysis; E.B. and H.-G.D. conducted the experiments and analysed the data; E.B. wrote the first draft; and all the authors commented and revised the article.

## Additional information

**Competing interests:** The authors declare no competing financial interests.

