## [Peer Review File · Nature Communications]

Reviewers' comments:

Reviewer #1 (Remarks to the Author):

The paper "Fronts and waves of cellular actin polymerization: the bistable mechanism of circular dorsal ruffles" combines experimental studies with theoretical modeling to the biological phenomenon of circular dorsal ruffles. Circular dorsal ruffles are important for the vesicular uptake of extracellular matter and thus gained a fair degree of attention from biologists. Thus the authors have investigated an important biological phenomenon, and the synergy between experiments and theoretical modeling constitutes a strength for the paper. Overall I believe this a worthwhile paper and deserves publication. Nevertheless there is an important question or concern that I would like to draw attention to. The authors state "... the phenomenology of initial expansion and succeeding contraction of wavefronts to localized points, ..., is unique and currently neither understood nor captured by any existing modeling attempt." This I believe is not entirely accurate, there have been recent papers on actin waves that could capture important aspects of such waves. Two such examples are: 1) V. Khamviwath, J. Hu, and H. G. Othmer, A Continuum Model of Actin Waves in Dictyostelium discoideum, PLoS ONE 8, e64272 (2013), and 2) Vaibhav Wasnik and Ranjan Mukhopadhyay, Modeling the dynamics of dendritic actin waves in living cells, Phys. Rev. E 90, 052707 (2014). While neither of the models developed in the two papers are particularly targeted towards studying circular dorsal ruffles, both demonstrate circular expanding waves and 1) in particular also discusses retracting waves. While the two models bear resemblance to the model proposed in the current paper, there are also important differences. Most importantly, bistability does not appear to be crucial to front propagation in either 1) or 2). Nevertheless reference 1) shows an enrichment of F-actin interior to the wavefront. This raises the question about whether the current study demonstrates conclusively the existence of bistability or whether it demonstrates that a model based on bistability can explain the experimental results but retaining the possibility of alternative mechanisms that might be compatible with the experimental results. Thus I believe that the authors should discuss this issue and the relationship between the model proposed here and models such as those proposed in 1) and 2). It will be particularly valuable if the authors could discuss the issue of how to experimentally between such alternate models/mechanisms of actin front propagation.

Reviewer #2 (Remarks to the Author):

A model of macropinocytosis is proposed based on three actin states that differ in mobility and turnover rates: branched actin at the dorsal cell membrane, immobile actin in the cell cortex and stress fibers; and G-actin. I would not suggest publication of the paper in Nature Comm., because I feel the factual basis for the model is too weak and the model does not apply to the three-dimensional organization of a macropinosome.

Bistability of the actin system has repeatedly been modeled in recent papers, which might be of interest in the present context: see Lomatin et al., Nat. Cell Biol. 17, 1435 (2016); Byrne et al., Cell Systems 2, 38 (2016).

Specific Comments:

p. 2 and p. 10: The authors express the view that a lack of CDRs and failure of macropinocytosis facilitates the uncontrolled growth of tumor cells. On the opposite, inhibition of macropinocytosis has been shown to compromise the growth of Ras-transformed tumor cells (see, for instance, Commisso et al., Nature 497,633 (2013)).

p. 2: Hepatitis is a disease rather than a pathogen.

p. 4: Motile actin forms clusters on the wave front that move with velocities of $0.18 \mu\text{m} \cdot \text{s}^{-1}$. How are these clusters moved if no myosin is involved?

p. 5: "We lump up the activity of Wrap1 and PIP3 ... control protein I." PIP3 is not a protein. On

p.6, the diffusion of PIP3 is neglected, although it can diffuse in the membrane.

p.5: Roles of PIP3 and PIP2. No evidence for an inhibitory function of PIP3 is provided. Why is the role of PIP3 in signaling through the Rac-WAVE pathway not taken into account? PIP2 is usually in large excess over PIP3. If PIP2 is uniformly distributed, why should it not act in the interior region?

p. 5: Is gelsolin a capping or a severing protein?

p.6: How is the directed movement of actin clusters at the wave front represented in the model? The model appears to encompass only a diffusion term.

p. 6: The model is two-dimensional, whereas macropinocytosis is an intrinsic three-dimensional process since a volume of fluid has to be enclosed. The model may therefore account for bistability of actin organization underlying a plane membrane rather than for engulfment of a vesicle.

p. 6: Are no formins involved in macropinocytosis?

p. 8: In the model, reversal of the direction of propagation occurs when the front hits the boundary. Is this also true for the formation of real macropinosomes; do they always enter the cell boundary in order to get closed?

p. 9: The authors propose that no myosin is involved in cup closure. I would not expect myosin II to be involved; but what about one of the myosins I that are known to act in phagocytosis and macropinocytosis? MyoIE has been shown to associate with macropinocytic cups (Brzeska et al., Cytoskeleton (2016) 73(2):68-82).

Fig.2: Is the lower membrane in this scheme thought to be the ventral membrane of the cell? The dorsal ruffles should be lamellae in which two membranes envelop the cortical actin.

Günther Gerisch

Reviewer #3 (Remarks to the Author):

Circular dorsal ruffles (CDR) are actin-driven wave-like patterns that emerge at the dorsal (upper) side of cells. They were found to play a role in liquid uptake by macropinocytosis, and dysfunctions in their dynamics have been related to various pathogenic processes including cancer progression. However, convincing mathematical models that account for the rich wave dynamics involved in CDR formation are largely missing. In this manuscript, Bernitt et al. address this gap and propose a new mathematical description to explain the dynamics of CDRs based on a bistable reaction-diffusion system. They provide a full analysis of their model in the classical framework of nonlinear pattern forming systems and show a detailed comparison between their modeling results and experimental data. In particular, they convincingly demonstrate that their model can account for the typical sequence of CDR expansion and shrinkage as well as for several aberrant scenarios, such as the formation of pinned waves and spirals.

Overall, this is a very nice paper that combines traditional approaches of nonlinear pattern formation with a timely biological question in an original way and will prove highly useful for the dynamically growing community working on actin-driven wave phenomena in living cells.

I can recommend publication of this manuscript in Nature Communications provided the authors give convincing answers to the following points.

Model:

- Why is the inhibitory effect of I included in the form $1/(1+I)$? One could

also imagine terms such as "-IB" or other. Are there biological reasons for this? Or rather dynamical systems reasons, in the sense that other terms would not result in the desired dynamics? I, personally, consider also the latter type of argument as a very valid one but, anyway, the reasons for this choice should be explained and discussed in the text.

- If I understand correctly, the inhibitor I is assumed to be well-mixed inside the cell. Is this supported by what we know about the diffusive time scales of such molecules inside the cell? Please comment on this point.

- In Eq. (4), activation (cortical recruitment?) of the inhibitor is taken to depend only on B, not on F (or, perhaps better, on B+F?). Why is there no activation term depending on F? In Eq. (2,3) you assume that I does play a role also for the formation of F from G-actin.

- Please comment on the choice of your parameters as given in the Materials and Methods Section. How did you choose them? Was a lot of fine tuning required?

- General comment: Simple bistable systems typically have only one type of trigger wave that reverses its direction only in response to a change in parameters. Here, we have a bistable systems with two types of trigger waves. Is this a well-known situation from a dynamical systems point of view, or rather something very special? A short comment on this and a few references to the literature covering bistable reaction-diffusion systems could be useful here.

Comparison to experiments:

- The two types of wave fronts found in the model show different profiles. In particular, one of them has a pronounced dip in F+B in the wake of the peak (Fig. 4C). Also, the peak separation between F+B and I seems to be different for both types of waves. Are these features in agreement with experimental data? Please comment on this.

- The pinned wave front in the model emits secondary waves to the inside of the CDR (Fig. 6 and movie). Is this always present, or just in a specific regime? Was anything similar observed in experiments? In the movie from the simulation, this seems a very prominent feature that should at least be mentioned and briefly discussed in the text.

Minor points:

- page 7, line 174: Why is $D_B < 1$? Maybe give a short explanatory sentence.

- page 7, line 176: Shouldn't it be " $A = B + G + F$ "?

- Supporting Information, page 4, Fig. 2C: Why don't I see the red G-curve? Why is the I-curve not shown in any of these plots?

Response Letter: NCOMMS-16-23125

We thank the referees for their careful assessment of our work. Herewith we submit substantially revised version of our manuscript. We prepared two versions: One is the final version without any markups, while the other containing highlights with specific modifications:

- Referee 1: blue color
- Referee 2: green color
- Referee 3: purple color.
- Modifications in orange correspond to general minor corrections.

In addition, we added a black outline to the time stamps and the scale bar in Figure 1A to improve readability.

Response Referee #1

(original text by referee #1 in blue)

The paper “Fronts and waves of cellular actin polymerization: the bistable mechanism of circular dorsal ruffles” combines experimental studies with theoretical modeling to the biological phenomenon of circular dorsal ruffles. Circular dorsal ruffles are important for the vesicular uptake of extracellular matter and thus gained a fair degree of attention from biologists. Thus the authors have investigated an important biological phenomenon, and the synergy between experiments and theoretical modeling constitutes a strength for the paper. Overall I believe this a worthwhile paper and deserves publication.

Nevertheless there is an important question or concern that I would like to draw attention to. The authors state “... the phenomenology of initial expansion and succeeding contraction of wavefronts to localized points, ..., is unique and currently neither understood nor captured by any existing modeling attempt.” This I believe is not entirely accurate, there have been recent papers on actin waves that could capture important aspects of such waves. Two such examples are: 1) V. Khamviwath, J. Hu, and H. G. Othmer, A Continuum Model of Actin Waves in Dictyostelium discoideum, PLoS ONE 8, e64272 (2013), and 2) Vaibhav Wasnik and Ranjan Mukhopadhyay, Modeling the dynamics of dendritic actin waves in living cells, Phys. Rev. E 90, 052707 (2014). While neither of the models developed in the two papers are particularly targeted towards studying circular dorsal ruffles, both demonstrate circular expanding waves and 1) in particular also discusses retracting waves. While the two models bear resemblance to the model proposed in the current paper, there are also important differences. Most importantly, bistability does not appear to be crucial to front propagation in either 1) or 2). Nevertheless reference 1) shows an enrichment of F-actin interior to the wavefront. This raises the question about whether the current study demonstrates conclusively the existence of bistability or whether it demonstrates that a model based on bistability can explain the experimental results but retaining the possibility of alternative mechanisms that might be compatible with the experimental results. Thus I believe that the authors should discuss this issue and the relationship between the model proposed here and models such as those proposed in 1) and 2). It will be particularly valuable if the authors could discuss the issue of how to experimentally between such alternate models/mechanisms of actin front propagation.

We thank the referee for the constructive comments to our manuscript and the general support. We have added the two references (Khamviwath et al: [28], Wasnik et al: [26] and several others) and elaborated throughout the text on their relation to our work (page 2 lines 69-77):

“Actin waves have been observed at the ventral cell side or the cell periphery before [17, 18, 19, 20, 21, 22, 23] and have been studied via reaction-diffusion models in the context of excitable [24, 25, 16], wave instable [26], and bistable [27, 28] dynamics, as well as in terms of actin-membrane

shape feedback [15, 29, 30]. However, the phenomenology that incorporates the initial expansion of a circular wavefront from a localized initiation spot, eventual contraction, and ultimate collapse of which the latter process underlies the endocytotic function of CDRs, is unique and currently neither understood nor captured (as a whole) by any existing modelling attempt.”

(page 4 lines 87-91):

“The spatiotemporal dynamics is robustly determined by bistability and by wave instability of one of the two states. Consequently, our rather simple model captures and generalizes many aspects of actin waves that have been modeled separately [19, 21, 23, 25, 26, 28] ranging from single waves to spatiotemporal chaos-type dynamics.”

and several other parts (page 6 lines 174-176, page 9 lines 239-241, page 11 lines 298-302, moreover adaption of page 13 lines 361-373 and page 14 lines 389-398).

As the referee correctly acknowledged, our framework is based on both bistability and wave instability and thus captures several qualitatively distinct phenomena, which cannot be captured altogether by the previously suggested models of actin waves. In fact, the work by Wasnik and Mukhopadhyay already highlights the limited validity of Khamviwath et al., although also Wasnik and Mukhopadhyay address only the traveling wave phenomenon that is not related to CDRs. Nevertheless, we fully agree that a more concrete comparison to these works clarifies our novelty and thus, made significant effort to elaborate on this in the revision.

We believe that this revision fully addresses the referee’s comment.

Response Referee #2

(original text by referee #2 in green)

A model of macropinocytosis is proposed based on three actin states that differ in mobility and turnover rates: branched actin at the dorsal cell membrane, immobile actin in the cell cortex and stress fibers; and G-actin. I would not suggest publication of the paper in Nature Comm., because I feel the factual basis for the model is too weak and the model does not apply to the three-dimensional organization of a macropinosome.

We feel our work was misunderstood in parts. In fact, there is a profound difference between CDRs, which are propagating waves of polymerizing actin that undulate the dorsal cell membrane, and macropinosomes, which are vesicles that are formed by the collapse of large, lamellar membrane protrusions. Macropinosomes are usually formed by membrane ruffles, however, these do not necessarily stem from CDRs, as macropinocytosis can also occur due to peripheral cell ruffling or even cell blebbing, which are fundamentally different from CDRs ([12]). In particular, CDRs do not always form macropinosomes, e.g., when they fail to close properly (besides examples in our manuscript see the video that we uploaded to <https://dl.dropboxusercontent.com/u/10957735/non-prop-clos-CDR.avi>). CDR-mediated macropinocytosis depends on the successful formation of a macropinocytotic cup, which is a ring-shaped structure of lamellar protrusions that can grow on a closing CDR. Therefore, properly closing CDRs can be seen as precursory structures for macropinocytosis. The fact that macropinocytotic cups, due to their geometry, go in hand with formation of very large macropinosomes, underlies the significance of CDRs for efficient endocytotic uptake, which is the main motivation for our work.

The subject of our work is the study the dynamics of CDRs, with a focus on the mechanism that allows these actin waves to propagate and to reliably close back to focal points. In contrast, the formation of the macropinocytotic cup, during the final stage of CDR collapse, is explicitly beyond the scope of our work, which we stated in the initial manuscript already very clearly (page 6, lines 167 - 168 in the initial manuscript):

“The model is two-dimensional, assuming planar and thin cells, as we do not aim to describe features such as the membrane deformation by CDRs”

(page 12, lines 305 - 308 in the initial manuscript):

“While the model focuses on the biochemical components, it currently ignores the actual growth of the vertical, cup-shaped protrusions [4, 39, 10, 34] that may introduce additional details, such as feedback between membrane shape and actin activity [24, 13].”

We agree that for a full understanding of macropinocytosis a three-dimensional model would be indispensable. We appreciate the comments by referee# 2 also, as they made us aware of the potential confusion that might arise when reading our work, due to the somewhat unlucky historic naming of “circular dorsal ruffles”. We therefore changed several parts of our manuscript of which the most important are (page 1, lines 16-19):

“To form large vesicles this endocytotic mechanism relies on the collapse and closure of precursory structures, which are actin-based, ring-shaped vertical undulations at the dorsal (top) cell membrane, a.k.a., Circular Dorsal Ruffles (CDRs).”

(page 4, lines 91-94):

“Yet, the model does not attempt to realistically address the final stage of macropinosome formation and engulfing, which requires the description of large scale membrane deformations, which is beyond the scope of this work.”

(page 2, lines: 43-49):

“Importantly, although CDRs provide the basis for the formation of macropinocytotic cups, they are not necessarily structures of high vertical extension themselves (compare, e.g., the scanning electron micrographs of CDRs in [5], where they appear as shallow undulations, with [4], in which they rather form circular arrays of extended, lamellar protrusions). Thus, the terminology “ruffle”, which has historical reasons [8], is somewhat inappropriate in this context”

Regarding the supposedly too weak factual basis of our work, we are not sure which parts are meant in particular. We do, however, reply to all points raised in the following.

Bistability of the actin system has repeatedly been modeled in recent papers, which might be of interest in the present context: see Lomatin et al., *Nat. Cell Biol.* 17, 1435 (2016); Byrne et al., *Cell Systems* 2, 38 (2016).

We are thankful for the suggestions and refer to these works in the revised manuscript (citations [49], [50]).

Specific Comments:

p. 2 and p. 10: The authors express the view that a lack of CDRs and failure of macropinocytosis facilitates the uncontrolled growth of tumor cells. On the opposite, inhibition of macropinocytosis has been shown to compromise the growth of Ras-transformed tumor cells (see, for instance, Commisso et al., *Nature* 497,633 (2013)).

Indeed the role of CDRs in cancer is far from trivial. We intended to highlight the controversy by pointing out that their absence can, on the one hand, foster uncontrolled growth ([9,7,3] in the initial manuscript), while, on the other hand, their presence can support cancer cell migration ([7] in the initial manuscript). We are thankful for the suggestion to cite the work by Commisso et al., which is now included in the manuscript ([11]) together with additional text (page 2, lines 55 - 60):

“Thus a lack of CDRs presumably facilitates the uncontrolled growth of tumour cells [10, 7, 3]. On the other hand, CDR-mediated macropinocytosis has been identified as an important mechanism of nutrient uptake especially in tumor cells [11] and, further, CDRs have been suggested to support cancer cell migration by softening of the cytoskeleton through disruption of stress fibers [7].”

p. 2: Hepatitis is a disease rather than a pathogen.

We are thankful for pointing out this mistake and modified the sentence accordingly by adding the word “viruses” (page 2, line 61).

p. 4: Motile actin forms clusters on the wave front that move with velocities of $0.18 \mu\text{m} \cdot \text{s}^{-1}$. How are these clusters moved if no myosin is involved?

In the framework of our model these clusters correspond to the small and fractured waves that result from the wave instability (see Fig. 5), i.e., rapidly polymerising and depolymerising clusters that move by actin turnover and diffusion of proteins, and thus do not require myosins to be motile.

We added additional text and cross references in the revised version of the manuscript to clarify this point (caption Figure 1, page 3, page 11 lines 317-319):

“We note that the proximity to a wave-instability regime may also explain the small motile actin clusters that we identified within CDR wavefronts (Figure 1B).”

p. 5: “We lump up the activity of Wrap1 and PIP3 ... control protein I.” PIP3 is not a protein.

In fact, PIP3 is a phospholipid, as we correctly stated already on page 5 in the lines 125-127 of the initial manuscript. To avoid any misunderstanding in the revised manuscript, we changed “protein” to “complex” when referring to the inhibitory field *I* (as we do not strictly refer to PIP3, but a complex of PIP3 with, e.g., Arap1).

On p.6, the diffusion of PIP3 is neglected, although it can diffuse in the membrane.

A similar point was also raised by referee #3. Indeed PIP3 can diffuse in two dimensions, and so should the complex *I*. As we agree that it appears unnatural to the reader to a priori neglect diffusion of the *I* field, in the revised manuscript we have now included a diffusion term for *I* in the equations, and afterwards explain why we study the limit where it can be neglected. Since the complex *I* will be of relatively large size, diffuse in the membrane, and its diffusion is in fact dependent in a non-trivial way on the binding and unbinding rates to, e.g., *F*, its effective diffusion is much slower than that of g-actin. The diffusion of *I* is slow, as for *B*, and will only act to smear slightly the shape of the fronts. We therefore prefer to study a simpler system that has less free parameters, without losing the main richness and qualitative nature of the dynamics. We added text in several parts of the revised manuscript for clarification (page 7 lines 198-207, page 9 lines 201-235, page 15 lines 420-430, in the SI text: page 3 lines 49-50).

p.5: Roles of PIP3 and PIP2. No evidence for an inhibitory function of PIP3 is provided. Why is the role of PIP3 in signaling through the Rac-WAVE pathway not taken into account? PIP2 is usually in large excess over PIP3. If PIP2 is uniformly distributed, why should it not act in the interior region?

PIP2 and PIP3 are involved in numerous regulatory pathways of processes of cytoskeletal rearrangement. Both, PIP2 and PIP3, take part in regulatory pathways that lead to the activation of the Arp2/3 complex ([35,36]). In the case of PIP3, this pathway involves WAVE, while in the case of PIP2 N-WASP is involved. As already stated in line 150 of our initial manuscript, the major nucleation factor of actin in CDRs is N-WASP, while WAVE appears to be negligible, as reported by Legg et al. ([32]). We therefore do not consider the Rac-WAVE pathway as fundamental to CDRs.

The pathways that lead to inhibition of actin polymerization in CDRs were already described in lines 120-146 of the initial manuscript. In short, Arap1 recognizes PIP3 and suppresses activation of Rac & CDC42 via Arf, leading to reduced actin nucleation. Moreover, PIP3 is produced at the expense of PIP2. Therefore PIP2-mediated dislocation of the capping activity of gelsolin is diminished in regions of high PIP3 concentration, i.e., the CDR interior. The corresponding pathways are clearly described in our manuscript and well elaborated in the cited literature. We therefore see no need to modify the text.

p. 5: Is gelsolin a capping or a severing protein?

In fact it is both ([38]). We improved the corresponding text to make this clear (page 6 lines 160, 163).

p.6: How is the directed movement of actin clusters at the wave front represented in the model? The model appears to encompass only a diffusion term.

We do not understand this question. The fragmented actin clusters (as opposed to a continuous actin ring) result from a wave instability of the system. Therefore their motility is the result of the propagation of non-linear waves of actin polymerization and depolymerization in our reaction-diffusion model. No further mechanism is necessary to explain their existence and their motility. Especially, the clusters are not directed in their movement (see, e.g., Movie S1).

p. 6: The model is two-dimensional, whereas macropinocytosis is an intrinsic three-dimensional process since a volume of fluid has to be enclosed. The model may therefore account for bistability of actin organization underlying a plane membrane rather than for engulfment of a vesicle.

Yes. This point was discussed already in the initial version of the manuscript very clearly (lines 305-308 in the initial manuscript):

“While the model focuses on the biochemical components, it currently ignores the actual growth of the vertical, cup-shaped protrusions [4, 39, 10, 34] that may introduce additional details, such as feedback between membrane shape and actin activity [24, 13].“

As stated above (response to the general part of the report by Referee #2), the focus here is not on the engulfment of a vesicle, but on the wavefront expansion and reversal of its propagation direction. There are no claims made otherwise. As stated in the answer regarding the general question of the three-dimensional nature of CDRs above, we try to make this even clearer in the revised manuscript (see quotes above in the response to the general part of the report by Referee #2).

p. 6: Are no formins involved in macropinocytosis?

There are about 55 proteins and other molecules known to localize to CDRs ([5-7]), but formins are, to the best of our knowledge, not among those. However, we note that even a potential involvement in macropinocytosis does not necessarily imply a function in the propagation mechanism of CDRs.

p. 8: In the model, reversal of the direction of propagation occurs when the front hits the boundary. Is this also true for the formation of real macropinosomes; do they always enter the cell boundary in order to get closed?

A macropinosome is the vesicle, which is eventually formed upon CDR collapse, so we like to point out that macropinosomes are not identical with CDRs. The reversal at boundaries is one of the most pronounced behaviours of CDRs. In fact, CDRs can also reverse, e.g., at the cell nucleus (see, e.g., the time lapse in Fig. 1 A, in which the CDR reverses at the nucleus. Also in Fig. 1D one CDR has a pronounced dent, which results from the fact that CDRs strongly avoid to cross the nucleus). As we point out in the SI (Text S4 “Counter propagating front solutions”), the wavefront bistability causes that even very small perturbations of an initial profile can lead to a switch between the evolution of the system from one solution to the other (demonstrated in Figure S2). We believe that in the heterogeneous environment of actual cells several scenarios can correspond to such a perturbation, including approach of the cell edge and also the nucleus. Since the cell boundary is the compartment of the cell where CDR reversal occurs most reliably, we chose to focus on the scenario of CDR reversal at this location.

p. 9: The authors propose that no myosin is involved in cup closure. I would not expect myosin II to be involved;

As has been already noted, we do not describe the final stages of cup closure and vesicle formation. When we use the terms “collapse” and “contract” we mean the inwards propagation of the CDR ring towards a single focal point. During this propagation there is no evidence for a role of myosin-II, see e.g., citation [15] in the manuscript.

but what about one of the myosins I that are known to act in phagocytosis and macropinocytosis? MyoIE has been shown to associate with macropinocytic cups (Brzeska et al., Cytoskeleton (2016) 73(2):68-82).

Indeed, Myosin I is found in CDRs (citation [5] in the manuscript). The work by Brzeska shows impressively that indeed the profiles of Myosin IF are very reminiscent to that of PIP3. However, as myosin Is are usually mostly involved in anchoring of the filaments to the membrane, it is not clear how they affect the reaction-diffusion mechanisms of actin polymerisation/depolymerisation. If the referee means that myosin-I could play a role during the large membrane deformations at the stages of vesicle formation, he/she may be right but we do not address the final stage of the macropinocytosis process in this work.

Fig.2: Is the lower membrane in this scheme thought to be the ventral membrane of the cell? The dorsal ruffles should be lamellae in which two membranes envelop the cortical actin.

Yes, the lower membrane in Fig.2 is indeed the ventral membrane of the cell, in contact with the substrate.

The three-dimensional shape of CDRs is a relatively complex issue (see Fig. R1). In fact, CDRs can grow lamellae of considerable extent, especially after the reversal of CDRs and towards their closure, i.e. upon formation of the macropinocytotic cup (SEM images in [4] and [12]). However, these rather complicated shapes are not required for the propagation of CDRs, as CDRs can also proceed to expand and collapse without forming extended lamellae ([5]). For this reason, and since we do not want to make things more complicated than they are, we decided to picture only a moderate membrane undulation in Fig. 2. We added new text (page 2, lines: 43-49):

“Importantly, although CDRs provide the basis for the formation of macropinocytotic cups, they are not necessarily structures of high vertical extension themselves (compare, e.g., the scanning electron micrographs of CDRs in [5], where they appear as shallow undulations, with [4], in which they rather form circular arrays of extended, lamellar protrusions). Thus, the terminology “ruffle”, which has historical reasons [8], is somewhat inappropriate in this context” and modified Figure 2 (addition of labels indicating dorsal/ventral membrane and substrate) to make this point more transparent.

Fig. R1: DIC micrographs of the same CDR at nearly identical time points at different vertical positions showing that CDRs can have both, regions of relatively flat undulation (3) and high lamellar protrusions (4) simultaneously. White rectangles mark the positions of the ROIs 1-4. The close-up views of the ROIs highlight details of the cell, verifying that the respective part of the cell is in focus. Scale bar top row: $10 \mu\text{m}$, scale bar ROIs: $1 \mu\text{m}$ (valid for all ROIs).

We hope that on the basis of our revisions and clarifications, the referee acknowledge the novelty of our results and reconsider his/her recommendation.

Response Referee #3

(original text by referee #3 in purple)

Circular dorsal ruffles (CDR) are actin-driven wave-like patterns that emerge at the dorsal (upper) side of cells. They were found to play a role in liquid uptake by macropinocytosis, and dysfunctions in their dynamics have been related to various pathogenic processes including cancer progression. However, convincing mathematical models that account for the rich wave dynamics involved in CDR formation are largely missing. In this manuscript, Bernitt et al. address this gap and propose a new mathematical description to explain the dynamics of CDRs based on a bistable reaction-diffusion system. They provide a full analysis of their model in the classical framework of nonlinear pattern forming systems and show a detailed comparison between their modeling results and experimental data. In particular, they convincingly demonstrate that their model can account for the typical sequence of CDR expansion and shrinkage as well as for several aberrant scenarios, such as the formation of pinned waves and spirals.

Overall, this is a very nice paper that combines traditional approaches of nonlinear pattern formation with a timely biological question in an original way and will prove highly useful for the dynamically growing community working on actin-driven wave phenomena in living cells.

I can recommend publication of this manuscript in Nature Communications provided the authors give convincing answers to the following points.

Model:

- Why is the inhibitory effect of I included in the form " $1/(1+I)$ "? One could also imagine terms such as " $-IB$ " or other. Are there biological reasons for this? Or rather dynamical systems reasons, in the sense that other terms would not result in the desired dynamics? I, personally, consider also the latter type of argument as a very valid one but, anyway, the reasons for this choice should be explained and discussed in the text.

We have now added the following explanation (page 7 lines 189-196) for motivating our choice of incorporation of the inhibitor into the equations:

"Since I is an inhibitor of actin polymerization rather than a degrading complex, the most natural choice to describe its effect on the polymerization reaction is via a simple rational function, akin to a Michaelis-Menten type term, that accounts for convergence to zero polymerization at high inhibitor concentration without allowing for negative polymerization rates [40]. Stress fibers in proximity to the dorsal membrane are similarly affected, and therefore also the polymerization reaction of F is inhibited by I ."

- If I understand correctly, the inhibitor I is assumed to be well-mixed inside the cell. Is this supported by what we know about the diffusive time scales of such molecules inside the cell? Please comment on this point.

Referee #2 raised a similar point (p6). In the revised manuscript we have now included a diffusion term for I in the equations, and afterwards explain why we study the limit where it can be neglected. Since the complex I will be of relatively large size, diffuse in the membrane, and its diffusion is in fact depending in a non-trivial way on the binding and unbinding rates to, e.g., F , its effective diffusion is much slower than that of g-actin. Incorporation of the diffusion of I into the model will only act to smear slightly the shape of the fronts. We therefore prefer to study a simpler system that has less free parameters, without losing the main richness and qualitative nature of the dynamics. We added text in several parts of the revised manuscript for clarification (page 7 lines 198-207, page 9 lines 230-235, page 15 lines 420-430, in the SI text: page 3 lines 49-50).

- In Eq. (4), activation (cortical recruitment?) of the inhibitor is taken to depend only on B , not on F (or, perhaps better, on $B+F$?). Why is there no activation term depending on F ? In Eq. (2,3) you assume that I does play a role also for the formation of F from G-actin.

The exact pathways that lead to activation of I from B are in fact unknown presently. However, the secondary wavefront of Arp1 behind the wavefront of polymerized actin in expanding CDRs is

very clear from the experimental data from [33] that we show in the SI (Figure S1). In contrast, in the F-dominated cell bulk, no Arap1 is found.

The inhibitory action of I on F and B is due to capping/severing of, e.g., gelsolin and capZ. Therefore, there is no reason to believe that the inhibition of F through I should imply a converse regulation.

We modified the corresponding text (page 5 lines 148-152) in our manuscript to make our motivation for the positive feedback from B on I more transparent:

“Both of these components have been shown to localize in CDR interiors; in the case of Arap1 it is in the form of a secondary wavefront that follows the wavefront of actin in expanding CDRs, implying a positive feedback from B on PIP3/Arap1 (SI Text, S1, Figure S1) [14, 33].”

- Please comment on the choice of your parameters as given in the Materials and Methods Section. How did you choose them? Was a lot of fine tuning required?

Our approach was based on a rough estimation of the relative time scales for, e.g., the diffusion constants. Moreover, we assumed the kinetic constants to be of similar magnitude. On this basis, the different regimes of the model could be identified easily via variation of the control parameter A . We added corresponding text to the manuscript (page 15 lines 420-430):

“After neglecting diffusion of I , the model in dimensionless form has six parameters, of which effectively four govern the dynamic regimes of the system (see SI Text, S3). Two-dimensional diffusion of B and relatively large particle sizes relative to the size of g-actin imply a small diffusion constant compared to the diffusion of unity for G . For the case of the complex I we expect small diffusion due to relatively large particle size and the coexistence of bound and unbound states to, e.g. F , that we do not explicitly model in our equations. We choose all kinetic constant to be of similar magnitude relative to each other and use the total actin concentration as a control parameter for the demonstration of the different regimes of the model.”

- General comment: Simple bistable systems typically have only one type of trigger wave that reverses its direction only in response to a change in parameters. Here, we have a bistable systems with two types of trigger waves. Is this a well-known situation from a dynamical systems point of view, or rather something very special? A short comment on this and a few references to the literature covering bistable reaction-diffusion systems could be useful here.

We added on page 10 the corresponding keyword and two related citations dealing with wavefront bistability, e.g. in the context of the Ising Bloch bifurcation (lines 266-269):

“Thus, the bistable nature of our model equations allows also the emergence of wavefront bistability, which is known in non-equilibrium systems, e.g., the Ising-Bloch front bifurcation [46, 47].”

Comparison to experiments:

- The two types of wave fronts found in the model show different profiles. In particular, one of them has a pronounced dip in $F+B$ in the wake of the peak (Fig. 4C). Also, the peak separation between $F+B$ and I seems to be different for both types of waves. Are these features in agreement with experimental data? Please comment on this.

Even though we do believe that this dip does indeed exist in weak form, the inhomogeneities in the F field of real cells make it very hard to make determined statements along these lines. We therefore decided not to compare this feature to the corresponding data. We included a corresponding statement on page 10 (lines 278-284):

“We note that a detailed comparison between the experimental and theoretical profiles is not given. The reason for this is that fluorescence imaging of CDRs gives the two-dimensional projection of three-dimensional structures, which can lead to the overestimation of the local actin density in case of significant vertical extension of CDRs. On this basis we currently cannot say with confidence if certain features, such as the pronounced dip in the observable actin in Figure 4C, are also found in experiments.”

Regarding the relative position of the maxima in the B and the I field of the extending and retracting fronts: the simulations do indeed agree with the experimental observation; there is a

corresponding description in the SI. We, however, failed to reference this in the main text in the initial manuscript. We added text to comment on the dip and inserted the missing reference (page 10 lines 266-272 in the main text):

“An intriguing outcome of our numerical results is the co-localization of the peaks of the B and the I field of the back-propagating front (Figure 4C), which appears to be in agreement with experimental observations (see also SI Text, S1, Figure S1).”

and page 1 lines 22-26 in the SI text:

“The relative positions of the maxima of polymerized actin and Arap1 are therefore different between expanding and contracting CDRs; note that for the expanding wavefront there is a pronounced peak of Arap1 following that of actin, whereas for the reversing wave both peaks co-localize, as indicated by the yellowish colors in Figure 1B. We found the same phenomenon in the results of our simulations (compare Figure 1B to Figure 4B and C in the main text).”

- The pinned wave front in the model emits secondary waves to the inside of the CDR (Fig. 6 and movie). Is this always present, or just in a specific regime? Was anything similar observed in experiments? In the movie from the simulation, this seems a very prominent feature that should at least be mentioned and briefly discussed in the text.

We added text to explain the dynamics of pinned CDRs (page 12, lines 330-339):

“Close inspection reveals that the oscillatory motion of the wavefronts at the boundary in Figure 6B results from periodic emission of wave trains from the boundary, which continue to propagate with weak amplitude towards the domain center. Although stalled CDRs do in fact perform oscillatory motion (Figure 6D), the experimental data show no evidence of actual wave trains. This might be due to their low amplitude that is possibly beyond experimental detectability. At even higher values of A no oscillations of stalled wavefronts are observed.”

Minor points:

- page 7, line 174: Why is $D_B < 1$? Maybe give a short explanatory sentence.

The reason is that the system is made dimensionless and the diffusion of g -actin is taken as one. Since the diffusion of the CDR f -actin is at/in the 2-d membrane, it has to be slower than one. We extended the corresponding sentence on page 8 (lines 202-204) to make it clearer:

“Compared to G , which is free to diffuse in three dimensions, B and I diffuse only at or in the two-dimensional dorsal cell membrane and thus have correspondingly small diffusion constants of $D_B, D_I < 1$.”

- page 7, line 176: Shouldn't it be " $A = B + G + F$ "?

We are very thankful for noticing of this typo, which we have corrected.

- Supporting Information, page 4, Fig. 2C: Why don't I see the red G -curve?

Why is the I -curve not shown in any of these plots?

We added a plot of the I -curve and changed the color of the G -curve. We verified that there was trouble displaying the G -curve on some computers due to an unknown artefact of vector graphics. We fixed the problem.

We believe that these revisions fully address the referee's comments.

REVIEWERS' COMMENTS:

Reviewer #1 (Remarks to the Author):

I have gone over the revised manuscript and response letter, and am convinced that my questions and concerns have been appropriately addressed. Thus I recommend acceptance of the manuscript for publication.

Reviewer #2 (Remarks to the Author):

The model presented focuses on the expansion and closure of dorsal ruffles in a 2-dimensional projection. However, the Introduction addresses at the very beginning macropinocytosis and volume control of fluid uptake, which are intrinsically 3-dimensional processes. The invagination of a membrane region is not covered by the model. I still believe, therefore, that in its present state the model fails in explaining the essentials of the process that renders the dorsal ruffles of general interest to a cell biology readership.

The draft should be carefully checked for misleading or inconsistent statements as, for instance, the following ones.

p. 8, line 210 says "The reaction kinetics of the model conserves the total amount of (local) actin concentration." Local conservation is unlikely to be realistic in the light of fast diffusing G-actin and an immobile polymerized state. It also appears to be inconsistent with the statement in line 218 about "bifurcation under variation of the total actin concentration". (By the way: What is the total amount of a concentration?)

p. 11, line 321: please clarify the meaning of "excitable pulses" reminiscent of results from ventral actin waves.

p. 14, line 380-382: The authors claim that the model contrasts to previous work attributing bistability to upstream GTPase switches. However, as pointed out in line 152, Arap1, introduced into the model as an inhibitor, acts by down-regulating upstream Rac and Cdc42.

p. 14, line 391 and elsewhere: "coexisting counter propagating fronts". I understand that the front either expands or, at another time, collapses. Or does "coexisting" mean something else?

I have two questions to equations (1)-(4) on p. 7. Why is the diffusion coefficient left out in equation (3)? In equation (3) it makes sense to attribute a plus sign to diffusion, because actin monomers are depleted. But why has diffusion a plus sign in equation (1) where the product is diffusing out of the site where B is generated?

Reviewer #3 (Remarks to the Author):

The authors have addressed all comments and suggestions of my previous report. I can now recommend publication in Nature Communications.

Response to Reviewer #2:

- “The model presented focuses on the expansion and closure of dorsal ruffles in a 2-dimensional projection. However, the Introduction addresses at the very beginning macropinocytosis and volume control of fluid uptake, which are intrinsically 3-dimensional processes. The invagination of a membrane region is not covered by the model. I still believe, therefore, that in its present state the model fails in explaining the essentials of the process that renders the dorsal ruffles of general interest to a cell biology readership.”

Response: Although our model captures several qualitatively distinct phenomena which are unique to CDRs, referee #2 (unlike the other two referees) feels that without membrane deformations our model is invalid. We disagree, since our observations and that of others (e.g., [5]) indicate that membrane deformations are rather small and persistent throughout most of the process of CDR expansion and collapse, except for the final stages where a vesicle is formed. Moreover, the reviewer does not propose any explicit example for the need (besides stating that it is missing) to include membrane deformations, i.e., when describing the phenomena of CDR dynamics. In any case, in accord with his previous report, we have double checked that there is no confusion in the text about the role of the membrane in our model. While we find the subject already clear, we have further emphasized this point in lines 185-190.

- “p. 8, line 210 says “The reaction kinetics of the model conserves the total amount of (local) actin concentration.” Local conservation is unlikely to be realistic in the light of fast diffusing G-actin and an immobile polymerized state. It also appears to be inconsistent with the statement in line 218 about “bifurcation under variation of the total actin concentration”. (By the way: What is the total amount of a concentration?)”

Response:

We improved the clarity of this sentence and the implications, see lines 224 and 233 in current revision.

- “p. 11, line 321: please clarify the meaning of “excitable pulses” reminiscent of results from ventral actin waves.”

Response: We have clarified this point, see lines 345-348 in the current revision.

- “p. 14, line 380-382: The authors claim that the model contrasts to previous work attributing bistability to upstream GTPase switches. However, as pointed out in line 152, Arap1, introduced into the model as an inhibitor, acts by down-regulating upstream Rac and Cdc42.”

Response: The referee is correct, in general. However, we do not claim that the pathways that are responsible for the regulation of actin polymerization in our

framework are unique nor do we deny that they involve well-known proteins such as Cdc42 and Rac. These Rho-GTPases are traditionally assumed to be switches that trivially suggest a bistable nature in actin regulation. In our work we show that bistability does not necessarily require molecular switches, as the mathematical structure of the reaction dynamics of actin can lead to the same phenomenon of bistability. Consequently, no changes here are required.

- “p. 14, line 391 and elsewhere: “coexisting counter propagating fronts”. I understand that the front either expands or, at another time, collapses. Or does “coexisting” mean something else?”

Response: We added text to the results part, where the term “counter propagating” appears for the first time (line 287) to avoid any confusion. Additionally we have added the word “distinct” in the discussion part (line 389).

- “I have two questions to equations (1)-(4) on p. 7. Why is the diffusion coefficient left out in equation (3)? In equation (3) it makes sense to attribute a plus sign to diffusion, because actin monomers are depleted. But why has diffusion a plus sign in equation (1) where the product is diffusing out of the site where B is generated?”

Response:

As stated in the lines 177-180 (directly before introduction of the model equations), the terms follow standard forms and the model is presented in its unit-less form. The reason for a diffusion coefficient of unity for G (eq 3) is that the whole system was scaled so that $D_g = 1$. This was stated in line 201 of the manuscript. Nevertheless, we have further modified the sentence (line 214) to improve the clarity.

Regarding the diffusion of B and the sign: the diffusion terms are presented in the simplest possible way, i.e., they are only dependent on the gradient of diffusive fluxes. Therefore, the positive signs of the diffusion terms are simply set by Fick's laws. No changes here are required.